

# Steroidal glycoalkaloids from *Solanum nigrum* target cytoskeletal proteins: an *in silico* analysis

Rumana Ahmad

Department of Biochemisty, Era's Lucknow Medical College and Hospital, Era University, Lucknow, Uttar Pradesh, India

## ABSTRACT

**Background**. *Solanum nigrum* (black nightshade; *S. nigrum*), a member of family Solanaceae, has been endowed with a heterogeneous array of secondary metabolites of which the steroidal glycoalkaloids (SGAs) and steroidal saponins (SS) have vast potential to serve as anticancer agents. Since there has been much controversy regarding safety of use of glycoalkaloids as anticancer agents, this area has remained more or less unexplored. Cytoskeletal proteins like actin play an important role in maintaining cell shape, synchronizing cell division, cell motility, etc. and along with their accessory proteins may also serve as important therapeutic targets for potential anticancer candidates. In the present study, glycoalkaloids and saponins from *S. nigrum* were screened for their interaction and binding affinity to cytoskeletal proteins, using molecular docking.

**Methods**. Bioactivity score and Prediction of Activity Spectra for Substances (PASS) analysis were performed using softwares Molinspiration and Osiris Data Explorer respectively, to assess the feasibility of selected phytoconstituents as potential drug candidates. The results were compared with two standard reference drugs doxorubicin hydrochloride (anticancer) and tetracycline (antibiotic). Multivariate data obtained were analyzed using principal component analysis (PCA).

**Results**. Docking analysis revealed that the binding affinities of the phyto-constituents towards the target cytoskeletal proteins decreased in the order coronin>villin>ezrin>vimentin>gelsolin>thymosin>cofilin. Glycoalkaloid solaso-nine displayed the greatest binding affinity towards the target proteins followed by alpha-solanine whereas amongst the saponins, nigrumnin-I showed maximum binding affinity. PASS Analysis of the selected phytoconstituents revealed 1 to 3 violations of Lipinski's parameters indicating the need for modification of their structure-activity relationship (SAR) for improvement of their bioactivity and bioavailability. Glycoalkaloids and saponins all had bioactivity scores between $-5.0$ and $0.0$ with respect to various receptor proteins and target enzymes. Solanidine, solasodine and solamargine had positive values of druglikeness which indicated that these compounds have the potential for development into future anticancer drugs. Toxicity potential evaluation revealed that glycoalkaloids and saponins had no toxicity, tumorigenicity or irritant effect(s). SAR analysis revealed that the number, type and location of sugar or the substitution of hydroxyl group on alkaloid backbone had an effect on the activity and that the presence of $\alpha$-L-rhamnopyranose sugar at C-2 was critical for a compound to exhibit anticancer activity.

Corresponding author
Rumana Ahmad,
rumana_ahmad@yahoo.co.in

**PeerJ** ___________________________________________________

**Conclusion**. The present study revealed some cytoskeletal target(s) for *S. nigrum* phytoconstituents by docking analysis that have not been previously reported and thus warrant further investigations both *in vitro* and *in vivo*.

## INTRODUCTION

Black nightshade (*Solanum nigrum)*, commonly known as Makoi in India, has been traditionally used in Southeast Asia, particularly the Indian subcontinent, as a panacea for several ailments (especially liver disorders) since time immemorial (*Li et al., 2009*; *Nawab et al., 2012*; *Wannang et al., 2008*; *Lee & Lim, 2003*; *Javed et al., 2011*; *Kang, Jeong & Choi, 2011*; *Lin et al., 2008*; *Hsieh, Fang & Lina, 2008*). *Solanum* sp. have been reported to possess a broad spectrum of activities viz. cytotoxic (*MahadevI et al., 2015*), antifungal (*Singh et al., 2007*), antiviral (*Arthan et al., 2002*), molluscicidal (*Silva et al., 2006*), antimalarial (*Makinde, Obih & Jimoh, 1987*), etc. Extracts of various plant parts of the genus have been shown to possess potent anticancer (*Patel et al., 2009*; *Raju et al., 2003*), antimicrobial (*Al-Fatimi et al., 2007*) and antiulcerogenic activities (*Jainu & Devi, 2006*). Recently, much attention has been directed towards the anticancer potential of the herb *S. nigrum* although the mechanism(s) for growth inhibitory and apoptosis inducing activity of *S. nigrum* have still not been properly understood or elucidated (*Gabrani et al., 2012*). *S. nigrum* is rich in SGAs (*Agarwal et al., 2010*; *Jain et al., 2011*) viz. solasodine, solanidine, alpha-solanine, solasonine, solamargine, diosgenin, solavilline and solasdamine (*Kuo et al., 2000*; *Liu et al., 2004*; *Chang et al., 1998*; *Huang, Syu & Jen-kun Lin, 2010*), most of which have been reported to possess possible antitumor properties though they have not been fully investigated (*Li et al., 2008b*). The cytotoxic activity of both crude extracts as well as isolated components has been evaluated against a panel of cancer cell lines viz. HepG2 (*Ji et al., 2008*), HT29 (*Lee et al., 2004a*), HCT-116 (*Lee et al., 2004b*), MCF-7 (*Son et al., 2003*), U14 (*Li et al., 2008a*), HeLa (*Oh & Lim, 2007*) as well as normal cell lines (*Moglad et al., 2014*) and on animal models of cancer.

The cytoskeleton is a remarkable system of specialized filaments that is highly developed in eukaryotic cells. The cytoskeleton is responsible for the spatial organization of the cells, their shape, motility and dynamic interaction with other cell types as well as their ability to grow, divide, and adapt to changing circumstances and environments. The cytoskeleton is composed of three types of fibres viz. intermediate filaments having a diameter (10 nm) intermediate between those of the two other principal elements of the cytoskeleton and mainly responsible for providing mechanical support and strength to the cell; actin filaments (about 7 nm in diameter) are responsible for the overall shape and motility of the cell whereas microtubules (about 25 nm in diameter) are involved in intracellular trafficking, signaling and transport. Apart from the above mentioned functions, the cytoskeleton also plays a critical part in mitotic assembly and disassembly, cell division,

cytokinesis and apoptosis. But these cytoskeletal filaments are unable to function on their own and require the presence of a large number of accessory and regulatory proteins that either bind these filaments to other components or to each other. This set of accessory proteins is critical for the controlled assembly and disassembly of the cytoskeletal filaments both spatially and temporally (*Franklin-tong & Gourlay, 2008*).

Recently, the role of actin and its binding proteins (ABPs) has been discovered in regulation and triggering of apoptosis (*Moss & Lane, 2006*) and, subsequently, appearance of apoptosis-related morphological characteristics like cell rounding, chromatin condensation, membrane blebbing and formation of apoptotic bodies. Growing evidence pinpoints to the involvement of actin cytoskeleton as both a sensor and mediator of apoptosis (*Desouza, Gunning & Stehn, 2012*). It has been found that inhibition of actin depolymerization (disassembly) by F-actin stabilizing drugs causes induction of apoptosis through intrinsic pathway, though exact mechanism has still not been elucidated (*Odaka, Sanders & Crews, 2000*; *Posey & Bierer, 1999*; *Kim et al., 2003*; *Cioca & Kitano, 2002*; *Rao et al., 1999*). However, there are also studies reporting actin depolymerization as an inducer of apoptosis (*Genesca, Sola & Hotter, 2006*). Thus, there is a possibility that actin dynamics (i.e., the rate of actin polymerization/depolymerization) is a key modulator of the apoptotic signal (*Morley, Sun & Bierer, 2003*) and its stabilization as well as destabilization can stimulate apoptosis in target cells. Different cell types have different G-actin (monomer) to F-actin (polymer) ratio (*Karpova, Tatchell & Cooper, 1995*; *Gibbon, Kovar & Staiger, 1999*; *Snowman et al., 2002*; *Atkinson, Hosford & Molitoris, 2004*) and this may explain the differential response of actin stabilization and destabilization towards apoptosis in different cell types and species. Since normal cells have been shown to maintain high levels of G-actin (*Atkinson, Hosford & Molitoris, 2004*), it is hypothesized that remodeling of actin cytoskeleton may enable cancer cells in evading normal apoptotic signaling.

The regulation of actin by ABPs and accessory proteins also plays an important role in apoptosis. F-actin maintenance is regulated by gelsolin, coronin-1 and cofilin.

The cofilin/actin depolymerizing factor (ADF) promotes depolymerization and severing from the actin filaments and thus may contribute towards an increase in G:F ratio (*Bamburg, 1999*). The active (dephosphorylated) form of cofilin has been demonstrated to be targeted to mitochondria resulting in cytochrome c release and triggering of apoptosis (*Chua et al., 2003*). Cofilin in association with actin has also been shown to inhibit apoptosis and has a short-term pro-survival role in neuronal cells but such a role has not been observed in other cell types and needs to be investigated (*Minamide et al., 2000*; *Ashworth et al., 2003*; *Bernstein et al., 2006*).

Ezrin is an actin-associated protein that functions in the extrinsic pathway of apoptosis by mediating interaction of actin with the CD95/FasL death ligand (*Parlato et al., 2000*). Phosphorylated form of ezrin interacts with membrane bound proteins via its N-terminal domain (FERM) while it is attached to the actin cytoskeleton at the C-terminus. It is hypothesized that binding of death ligand to the death receptors on the cell membrane of the target cell causes transduction of signal to the actin cytoskeleton thereby triggering apoptosis via the extrinsic pathway (*Algeciras-Schimnich & Peter, 2003*; *Algrain et al., 1993*).

Downregulation of ezrin expression causes CD95 mediated apoptosis in H9 stem cells; thereby the role of ezrin in apoptosis remains more or less ambiguous and elusive.

Gelsolin is an ABP belonging to a conserved superfamily of proteins that contain the conserved gelsolin-like domain and includes the protein villin, amongst others. It is a potent actin severing protein that caps the barbed end of F-actin in presence of calcium ions preventing further actin polymerization (*McGough et al., 2003*). The anti-apoptotitc and pro-survival role of gelsolin has also been recently demonstrated in pancreatic beta-cells (*Ohtsu et al., 1997*; *Yermen, Tomas & Halban, 2007*). Gelsolin may also promote cell survival by complexing with a recently discovered N-RAS (neuroblastoma RAS viral oncogene homolog) (*Keller et al., 2007*). Another recent finding has been related to high susceptibly of gelsolin knock-out mice to brain injury following ischemia which is attributable to gelsolin/actin mediated regulation of calcium channels (*Endres et al., 1999*). Overexpression of gelsolin in Jurkat cells (*Ohtsu et al., 1997*) and neuronal cells (*Harms et al., 2004*) has been shown to inhibit apoptosis by blocking the loss of mitochondrial membrane potential. Gelsolin gene silencing in Ras-mutated HCT116 colon cancer cells also led to apoptosis *via* caspase activation (*Klampfer et al., 2004*). This is due to the regulation of voltage-dependent anion channels (VDACs) that are present on mitochondrial membranes, thus preventing the release of proaopototic factors (*Granville & Gottlieb, 2003*). The anti-apoptotic activity of gelsolin resides in the G5 segment of domain located towards the C-terminal end of the protein (*Qiao & McMillan, 2007*) which has been shown to bind to and inhibit VDAC channels located on the mitochondrial membrane thereby preventing loss of membrane potential (*Qiao & McMillan, 2007*; *Qiao et al., 2005*). Gelsolin-phosphoinositide complex may also inhibit caspase-3 activity thereby affecting the executional phase of apoptosis (*Azuma et al., 2000*). Thus, it is hypothesized that gelsolin may play a protective role against apoptosis in certain cell types viz. neurons and cancer cells. This aspect needs further exploration as it may yield therapeutic intervention against cancer and has been a subject of key analysis in the present paper. Gelsolin is targeted and cleaved by caspases to yield a pro-apoptotic N-terminal fragment and an N-myristoylated anti-apoptotic C-terminal fragment (*Koya et al., 2000*; *Sakurai & Utsumi, 2006*). Thus, it is becoming increasingly evident that gelsolin is the 'hotspot' regulator at multiple points in the apoptotic pathway.

Villin is a 92.5 kDa, tissue-specific ABP associated with the actin core bundle of the brush border of the epithelium in vertebrates (*Friederich et al., 1999*) having functions similar to gelsolin. While the C-terminal domain is responsible for organizing and bundling of actin filaments in a $Ca^{2+}$ independent fashion, the core domain has been found to be responsible for effecting actin-capping, nucleation, and severing in a $Ca^{2+}$ and phosphatidylinositol-dependent fashion, *in vitro* (*Friederich et al., 1999*). Thus, villin may be involved in the organization and stabilization of the F-actin core bundle and due to its severing ability, villin may have a biologically important role to play in remodeling of actin cytoskeleton *in vivo* during physiological conditions viz. repair of damaged intestinal epithelium.

Coronin is a regulator of Arp2/3, an ABP which has an actin nucleating and branching function (*Cai et al., 2007*). Out of seven mammalian coronin proteins, coronin-1 is preferentially expressed in hematopoietic cells and serves a pro-survival function (*Uetrecht*

& Bear, 2006). It prevents F-actin formation (*Foger et al., 2006*) and also inhibits intrinsic pathway mediated apoptosis (*Uetrecht & Bear, 2006*). Therefore, inhibition of coronin-1 by potential compounds may cause apoptosis in cancer cells and serve as an effective therapeutic intervention.

Vimentin is a type III IF protein abundant in cells having mesenchymal origin as well as cultured and tumor cells. Expression of vimentin in tumor cells undergoing metastasis is considered a marker of epithelial to mesenchymal transition (EMT) (*Hay, 2005*; *Mendez, Kojima & Goldman, 2010*). Regulation of vimentin though phosphorylation is integral for both mitosis and cell architecture and motility in interphase, and promoting proper and complete separation of vimentin between the newly formed daughter cells (*Lowery et al., 2015*; *Gruenbaum & Aebi, 2014*; *Snider & Omary, 2014*; *Hyder et al., 2008*; *Yasui et al., 2001*). Cytokinesis involves disassembly of vimentin IFs as an integral step. Depolymerization of IF is induced by phosphorylation of Ser[55] of vimentin by Cyclin-dependent kinase 1 (Cdk1) (*Tsujimura et al., 1994*; *Yamaguchi et al., 2005*; *Chou et al., 1990*). It is hypothesized that *S. nigrum* glycoalkaloids might exert their anticancer effect by interfering with intermediate filament disassembly; thus, leading to apoptosis. Also vimentin inhibition may prevent EMT, a critical event in breast cancer metastasis.

The β-thymosins are a family of conserved small MW (5 kDa) polypeptides that specifically bind monomeric G-actin, preventing its polymerization to F-actin (*Huff et al., 2001*). Overexpression of thymosin β10 (TB10) causes apoptosis in ovarian tumor cells (*Rho et al., 2005*). TB10 and actin bind E-tropomodulin (E-Tmod) at the same site. E-Tmod caps actin on the pointed end and is responsible for altering the actin filament length. E-Tmod overexpression has been shown to block TB10-mediated apoptotic activity, supporting the hypothesis that an inter-related dynamic equilibrium exists between TB10 and E-Tmod for regulating actin-mediated apoptosis (*Rho et al., 2005*; *Woo et al., 2012*).

Withanolides from the medicinal plant *Withania somnifera* are known to target the intermediate filament protein vimentin using molecular docking (*Bargagna-Mohan et al., 2007*; *Bargagna-Mohan et al., 2010*). Because of the similarity of structural backbone between the withanolides and alkaloids of *S. nigrum*, it was worthwhile to check the binding modes of major active components of *S. nigrum* with cytoskeletal and other actin binding proteins; hence the present study.

## MATERIALS AND METHODS

### Ligand Preparation

The selected phytoconstituents belonged to the class of steroidal aglycones (solanidine and solasodine), glycoalkaloids (solanine, solamargine, solasonine) and steroidal saponins (degalactotigonin, uttroside B and nigrumnin-I). The criteria for selection were their previously reported structure activity relationships (SARs) (*Huang, Syu & Jen-kun Lin, 2010*; *Atanu, Ebiloma & Ajayi, 2011*; *Li et al., 2008a*; *Li et al., 2008b*; *Zhou et al., 2006*), their demonstrated antitumor effects on various tumor cell lines (*Jain et al., 2011*; *Li et al., 2008a*; *Li et al., 2008b*; *Atanu, Ebiloma & Ajayi, 2011*; *Zhou et al., 2006*; *Chauhan et al., 2012*; *Nath et al., 2016*) and prospective targeted metabolic pathways (*Huang, Syu & Jen-kun Lin,*

*2010*; *Nath et al., 2016*). PubChem and ChEMBL databases were used for retrieval of 3D structures of the eight phytoconstituents and two reference drugs in SDF format. Before docking, energy minimization of ligands was performed by Merck Molcular Force Field (MMFF94). All ligand structures underwent structural optimization using AutoDock Tools (ADT) version 4.2.6.

## Target protein preparation

The 3D crystal structures of prospective receptors/protein targets (whose X-ray diffraction structures are available in RCSB database) of phytoconstituents were downloaded from Protein Data Bank (http://www.rcsb.org/pdb) in PDB format. All protein structures were subjected to refinement and energy minimization before docking analyses. The refinement was performed by addition of missing atoms in missing residues, addition of polar hydrogen atoms and Kollman charges, removal of crystallographic water molecules and external and extraneous ligands and ions from the protein. The structures were visualized in Accelrys Biovia Discovery Studio version 2017 R2 (Biovia, San Diego, CA, USA). Receptor energy minimization was carried out by using default constraint of 0.3Å RMSD (root mean square) and AMBER force field 14SB using Chimera 1.12. Energy calculations were made after structural refinement and removing structural inconsistencies. Minimization routine was performed by MMTK which is included with Chimera (*Pattersen et al., 2004*). The PDB IDs of the target proteins were as follows: vimentin (PDB ID: 1GK4), gelsolin (PDB ID: 3FFN), villin (PDB ID: 3FG7), ezrin (PDB ID: 4RMA), cofilin-1 (PDB ID: 4BAX).

## Homology modeling

Obtaining 3D structures of proteins is one of the first steps in the process of *in silico* docking. Since 3D structures of human coronin-1A and thymosin beta-4 were not available in the Protein Data Bank (http://www.rcsb.org), the same were generated using homology modeling. The term homology modeling, also known as comparative modeling and template-based modeling (TBM), refers to modeling a protein 3D structure using a known experimental structure of a homologous protein (the template). The amino acid sequences of human coronin-1A and thymosin beta-4 were obtained from universal protein knowledge database (UniProtKB) having IDs P31146 and P62328, respectively (Fig. 1). For sequence alignment, the sequences of COR1A and TYB4 were retrieved from different organisms in FASTA format using Expert Protein Analysis System, or ExPASy Molecular Biology Server in Geneva, Switzerland. Many applications require the amino acid sequence to be in FASTA format. The FASTA format includes the amino acid sequence in one-letter code, usually with 60 letters/line (*Lopez et al., 2007*). To run the alignment, additional sequences were chosen from distantly related organisms (3-4 bacterial and 3-4 eukaryotic), in order to get a better idea on the conservation of the most important amino acids. Care was taken to include the sequences marked 'reviewed' and the protein sequences from *Homo sapiens*. The sequences of COR1A and TYB4 (or their respective UniProt entry codes) were pasted into the template identification window of the Swiss Model server and were subjected to BLAST (Basic Local Alignment Search Tool) (*Altschul et al., 1997*) against the sequences of selected known protein structures from the ExPDB,

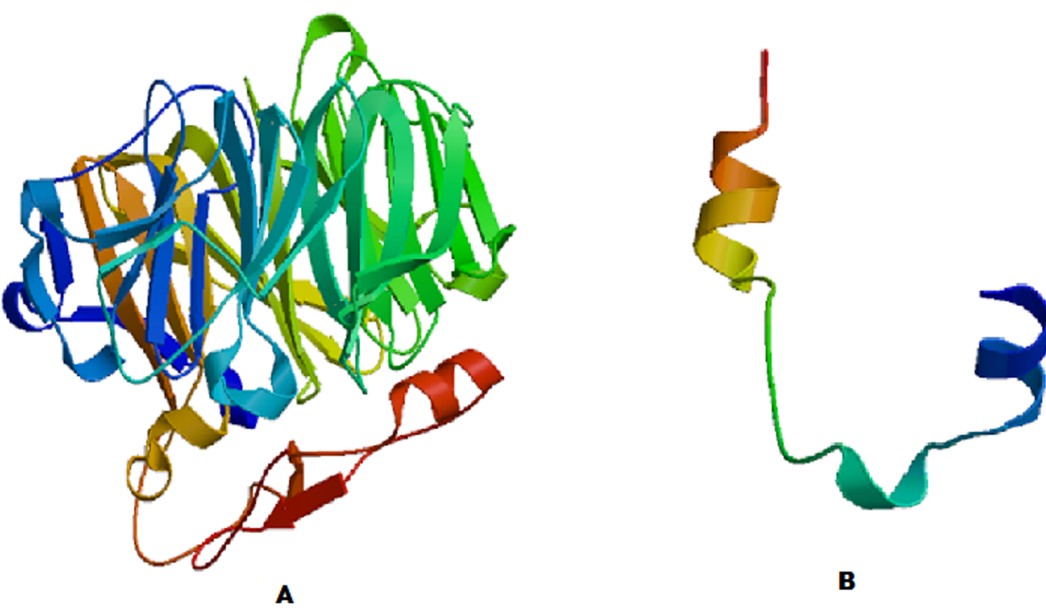

**Figure 1** Modeled structures of (A) COR1A and (B) TYB4.

the Swiss Model template library (SMTL version 2017-09-21, PDB release 2017-09-15). Modeling was done using Swiss Model server, which is relatively fast and provides good model quality assessment. There are several other online servers that may be used for modeling (like Phyre, I-TASSER or ROBETTA), which use more sophisticated algorithms, and can take days (or even weeks) to return the modeling request. Generally, a percentage sequence identity above 50% means a relatively straight forward modeling project, while anything below requires careful planning.

The query proteins COR1A and TYB4 contained 461 and 44 amino acids respectively. 3D atomic models of the query proteins were generated using ProMod3 Version 1.0.2 (http://swissmodel.expasy.org/). Three dimensional models of COR1A and TYB4 were visualized in Accelrys Biovia Discovery Studio version 2017 R2 (San Diego, CA, USA).

## Structure validation and quality analysis of homology models

The obtained homology models COR1A and TYB4 were subjected to quality analysis using Molprobity (http://molprobity.biochem.duke.edu/), an online tool for all-atom contacts and structure validation for proteins and nucleic acids (*Davis et al., 2007*; *Chen et al., 2010*). The web service is dependent on optimized hydrogen placement and all-atom contact analysis which is complemented by updated versions of covalent-geometry and torsion-angle criteria. MolProbity performs a few local corrections in automated manners that are presented in chart and graphical forms in order to support manual rebuilding. The validation analysis was performed through addition of H-atoms, all-atom contact analysis, Ramachandran and rotamer analyses and covalent-geometry analyses.

## Molecular docking of *solanum* phytoconstituents with selected targets

Phytoconstituents from *S. nigrum* were subjected to molecular docking analysis with selected protein targets using AutoDock version 4.2.6. In case of human coronin-1A and human thymosin beta-4, whose 3-D structures were generated using homology modeling and refined by Molprobity; the analysis of bound protein-ligand complexes was done using three molecular docking softwares viz. AutoDock version 4.2.6, AutoDock vina and iGEMDOCK version 2.1. All protein targets were also docked with their known physiological ligands as reported in literature (*Komura, Ise & Akaike, 2012*; *Janmey et al., 1987*; *Janmey & Stossel, 1987*; *Janmey & Matsudaira, 1988*; *Janmey et al., 1992*; *Niggli et al., 1995*; *Lin et al., 1997*; *Tsujita et al., 2010*).

### AutoDock

AutoDock is a free, automated software developed by Molecular Graphics Lab, Scripps Research Institute, La Jolla, CA 92037, USA, and is used for prediction of binding interactions between ligands and their biochemical targets (*Morris et al., 2009*). Minimum binding energy for a given ligand to its target protein is calculated by the software by exploring all available degrees of freedom (DOF) for the system. AutoDock version 4.2.6. employs the Lamarckian genetic algorithm and empirical scoring function. A free energy force field is used to evaluate conformations during docking simulations by the software in two stages. Firstly, the ligand and its target protein are unbound followed by energy evaluations in their bound state by the software. In the present study, rigid docking studies were performed by setting the angle of rotation to 7.5 degrees. AutoDock version 4.2.6 provided both rigid (no torsional flexibility to a protein as well as ligand) and flexible (torsional flexibility to a ligand with a rigid protein) docking.

### AutoDock vina

AutoDock Vina is again a freeware designed to be approximately two orders of magnitude faster compared to AutoDock 4, and at the same time more accurate in binding mode predictions (*Trott & Olson, 2010*). Faster results are obtained as a consequence of multithreading on multi-core machines. As opposed to AutoDock 4, AutoDock Vina automatically calculates the grid maps and clusters the results in a way transparent to the user.

### iGEMDOCK

iGEMDOCK version 2.1, is a graphical, automated software developed by the Institute of Bioinformatics, National Chiao Tung University, Taiwan, for integrated docking, screening and post-analysis (*Yang & Chen, 2004*). For docking, preparation of protein and ligand files was done. With the help of the software, binding sites for the ligand were defined and prepared. iGEMDOCK uses a generic evolutionary method (GA) to compute a ligand conformation and orientation relative to the binding site of protein target. Therefore, docking performance is directly related to the selected GA parameters. The selected GA parameters were as follows: population size = 800, generations = 80, number of solution = 10 and docking function as 'slow docking'. After generation of a set of poses, iGEMDOCK

recalculates the energy of each pose. The interaction data includes individual and overall energy terms. Best fit is chosen which represents the total energy of a predicted pose in the binding site of the protein. iGEMDOCK has an empirical scoring function given by:

$$\text{Fitness} = \text{vdW} + \text{Hbond} + \text{Elec}$$

where the vdW term is van der Waals energy, Hbond and Elect terms stand for hydrogen bonding energy and electrostatic energy, respectively.

## Analysis of docked ligand-protein complexes

The best orientations of the ligand-protein interactions were analyzed in Accelrys Biovia Discovery Studio version 2017 R2. Protein binding sites were predicted using 'find cavities' in the receptor site parameter of the tool. The docking software calculated the binding energies and $K_d$ values of phytoconstituents to each target protein. Lower binding energies and $K_d$ values meant greater affinity of the phytoconstituent towards a particular target protein.

## PASS analyis

PASS (Prediction of Activity Spectra for Substances) is an online web tool hosted at http://195.178.207.233/PASS/index.html. The software predicts biological activities of chemical compounds, including phytochemicals, on the basis of structure–activity relationship with a known chemical entity. The tool makes predictions on the pharmacological activity, mechanism of action as well as side effects such as carcinogenicity, mutagenicity, embryotoxicity and teratogenicity (*Parasuraman, 2011*; *Mathew, Suresh & Anbazhagan, 2013*; *Filimonov et al., 2014*; *Jamkhande, Pathan & Wadher, 2016*). In the the present study, PASS analysis was performed using OSIRIS Property Explorer version 4.51 (http://www.openmolecules.org/propertyexplorer/index.html).

## Lipinski's rule-of-five

Lipinski's rule of five (RO5) is used to evaluate druglikeness of a chemical compound possessing properties that would make it a likely or potential drug in humans (*Lipinski et al., 2001*). The oral activity of a drug compound is predicted by calculating certain molecular parameters like log P (partition coefficient), polar surface area, number of hydrogen bond donors, number of hydrogen bond acceptors and molecular weight. The rule states that most drug candidates with good membrane permeability must have $\log P \leq 5$, number of hydrogen bond acceptors $\leq 10$, and number of hydrogen bond donors $\leq 5$. In general, an orally active drug has no more than one violation of the given criteria.

Lipinski's rule of five is helpful in describing molecular properties of drug compounds required for estimation of important pharmacokinetic parameters such as absorption, distribution, metabolism and excretion. The rule is helpful in drug design and development (*Lipinski et al., 2001*; *Ertl, Rohde & Selzer, 2000*; *Veber et al., 2002*).

## Bioactivity score prediction

Drug score values indicate overall potential of a prospective compound to be a drug candidate. Molinspiration version 2016.10, a web-based tool, was used to predict the

bioactivity score of the phytoconstituents with respect to human receptors like GPCRs, ion channels, kinases, nuclear receptors, proteases and enzymes (*Proudfoot, 2002*). If the bioactivity score is found to be more than 0.0, then the compound is active; if it is between −5.0 and 0.0, then the complex is moderately active and if the bioactivity score is less than −5.0, then it is inactive (*Verma, 2012*).

**Toxicity potential assessment**

Toxicity risk assessment gives an idea about the probable side effects of compounds that may be used for further processing in drug development and discovery. The mutagenic, tumorigenic, irritant and reproductive toxicities were measured by means of pre-computed set of structural fragments. The prediction of different properties of molecules in the early stage is a vital step in drug discovery and development process. Toxic parameters of the phytoconstituents were generated and analyzed by the OSIRIS Data Warrior software version 4.5.1 (http://www.openmolecules.org/datawarrior/).

**Principal component analysis (PCA)**

PCA was performed using Osiris Property Explorer 4.5.1 for defining and visualizing various multidimensional "property spaces" by assigning dimensions to numerical descriptors of molecular structures of complexes and standard drugs viz. MW, % Absorption and TPSA (*Veber et al., 2002*; *Ghose & Crippen, 1987*). The bar charts and 3D scatter plots of principal components to depict druglikeness of the phytoconstituents versus standard drugs were made in OSIRIS Property Explorer 4.5.1 (http://www.openmolecules.org/propertyexplorer/index.html) and Accelrys Biovia Discovery Studio version 2017 R2, respectively.

# RESULTS

**The overall MolProbity score of homology models COR1A and TYB4**

MolProbity score provides a single number that represents the overall protein quality statistics. MolProbity score combines the clashscore, rotamer and Ramachandran evaluations into a single score, normalized to be on the same scale as X-ray resolution. Currently, the percentile scores are given for clashscore and for MolProbity score relative to the cohort of PDB structures within 0.25 Å of the file's resolution. Figures 1A and 1B depict the modeled 3D structures of human coronin 1-A (COR1A) and thymosin beta-4 (TYB4) respectively. In the present study, the molprobity scores of COR1A and TYB4 (Fig. 1) were determined to be 0.91 and 0.86 respectively, which were well within the allowed range (Tables S1, S2). Ramachandran plots of the modeled COR1A revealed that 95% residues lay inside the 'favored 98% contour' (Fig. S1) and there were two out of 393 Ramachandran outliers (Gly 129 and Val 229). In case of TYB4, 100% residues lay within the 'favored 98% contour' (Fig. S2) and there were 0 out of 34 Ramachandran outliers. Ramachandran plots for Gly, Pro and pre-Pro residues have also been shown for both COR1A and TYB4. In case of COR1A, the $C^\beta$ deviation kinemage showed each residue's $C^\beta$ position relative to an ideal $C^\beta$ and its three bond vectors (gray lines, Fig. S3). Circles mark the deviation distances, with the yellow circle at the 0.25 Å cutoff for outliers. Most of the distribution was found to be good, but Gly 129 and Val 229 formed part of an outlier cluster and

probably reflect distortions caused by a local fitting problem. In the case of TYB4, there were no outliers, and all residues lay within the yellow circle at the 0.25 Å cutoff for outliers (Fig. S3).

## Docking Studies of *S. nigrum* ligands with target proteins

Table 1A depicts the docking results of major components in *S. nigrum* extract(s) with respect to different cytoskeletal proteins. Calculated binding energies and dissociation constants ($K_d$) of the phytoconstituents with respect to different target proteins have been summarized in Table 1. Tables 1B and 1C respectively depict the various binding and energy parameters of different ligands with respect to coronin-1A and thymosin beta-4, predicted using AutoDock version 4.2.6, AutoDock vina and iGEMDOCK version 2.1. It is apparent from Tables 1B and 1C that in most cases, the kinetic and binding parameters did not vary significantly in presence or absence of structural refinement of coronin-1A and thymosin beta-4, afforded by Molprobity analysis. Table 2, parts a, b and c and Figs. 2 and 3 respectively depict the best docking poses of binding of steroidal alkaloids and their glycosylated derivatives (Fig. 2) and steroidal saponins (Fig. 3) to target proteins (the hydrogen bonds are shown as green dotted lines). It is apparent from Tables 1 and 2 that most phytoconstituents displayed potent binding to the protein coronin-1A and thymosin beta-4. The binding sites of the phytoconstituents on coronin-1A and thymosin as well as the interacting amino acids were predicted to be almost the same by all three molecular docking softwares (Tables 2B, 2C). For the best docking pose(s) of all eight phytoconstituents to coronin-1A and thymosin beta-4, generated using AutoDock version 4.2.6, AutoDock vina and iGEMDOCK version 2.1., refer to Figs. S4 and S5.

The cytoskeletal proteins were also studied for their binding to a number of physiological ligands (Table 3). The binding modes and interactions of various physiological ligands to their respective target protein(s) were studied using three different molecular docking softwares viz. AutoDock version 4.2.6, AutoDock vina and iGEMDOCK version 2.1. As reported in literature (*Strelkov et al., 2002*; *Komura, Ise & Akaike, 2012*), N-acetyl glucosamine displayed good binding to vimentin ($K_d$ 15.46 μM). Gelsolin displayed strong binding to polyphosphoinositides phosphatidylinositol 4,5-bisphosphate (PIP$_2$; $K_d$ 43.43 μM) and phosphatidylinositol 4-monophosphate (PIP; $K_d$ 4.39 μM) by AutoDock vina and iGEMDOCK. This too, was in agreement with previously reported studies (*Janmey et al., 1987*; *Janmey & Stossel, 1987*; *Lin et al., 1997*). Calcium also displayed binding to Gelsolin ($K_d$ 108.84 mM), as reported in literature (*Nag et al., 2009*). PIP$_2$ and PIP also displayed potent affinity towards villin with dissociation constants of 39.91 μM and 10.02 μM, respectively (and this was found to be in agreement with previously cited literature (*Janmey & Matsudaira, 1988*; *Janmey et al., 1992*). In contrast, villin also displayed affinity towards small physiological ligands such as acetate ($K_d$ 6.0 mM) and sulfate ($K_d$ 4.75 mM) (*Vermeulen et al., 2004*). Again, both PIP$_2$ and PIP showed good binding to human Coronin-1A having dissociation constants of 76.59 μM and 15.3 μM, respectively towards coronin (*Tsujita et al., 2010*). It was found that in all cases, PIP exhibited greater affinity for the protein target(s) viz. gelsolin, villin and coronin than the relatively bulkier PIP$_2$ which had lesser affinity due to the presence of an extra phosphate group. Coronin also displayed

**Table 1** Binding energies and dissociation constants ($K_d$) of *S. nigrum* phytoconstituents with respect to target protein(s).

**A. Binding energies (kcal/mol) and dissociation constants ($K_d$) towards selected target protein(s)**

| S. No | Phytoconstituent name | Vimentin B.E. | Vimentin $K_d$ | Gelsolin B.E. | Gelsolin $K_d$ | Villin B.E. | Villin $K_d$ | Ezrin B.E. | Ezrin $K_d$ | Cofilin-1 B.E. | Cofilin-1 $K_d$ |
|---|---|---|---|---|---|---|---|---|---|---|---|
| 1 | Solanidine | −7.59 | 2.71 μM | −8.08 | 1.2 μM | −8.63 | 468.76 nM | −8.43 | 664.78 nM | – | – |
| 2 | Solasodine | −8.18 | 1.0 μM | −7.52 | 3.07 μM | −9.51 | 106.05 nM | −7.25 | 4.89 μM | – | – |
| 3 | Alpha-Solanine | −7.12 | 5.99 μM | −6.66 | 13.18 μM | −7.87 | 1.7 μM | −8.16 | 1.05 μM | – | – |
| 4 | Solasonine | −7.82 | 1.87 μM | −6.8 | 10.38 μM | −5.84 | 52.18 μM | −7.84 | 1.8 μM | – | – |
| 5 | Solamargine | −6.95 | 8.1 μM | −6.09 | 34.2 μM | −6.45 | 18.8 μM | −7.25 | 4.83 μM | – | – |
| 6 | Degalactotigonin | −1.83 | 45.68 μM | −2.93 | 7.12 mM | −1.77 | 50.65 μM | −1.82 | 46.01 mM | −2.0 | 33.98 mM |
| 7 | Nigrumnin-I | −3.81 | 1.61 mM | −3.63 | 2.19 mM | −5.97 | 42.35 μM | −6.39 | 20.62 μM | – | – |
| 8 | Uttroside B | −1.89 | 40.84 mM | −2.76 | 9.51 mM | −3.77 | 1.72 mM | −2.76 | 9.5 mM | – | – |

**B. Energy parameters with quality analysis (Molprobity) using three docking softwares**

**Human Coronin-1A**

| S. No | Phytoconstituent name | AutoDock v4.2.6 B.E. (kcal/mol) | AutoDock v4.2.6 $K_d$ | AutoDock vina B.E. (kcal/mol) | AutoDock vina $K_d$ | iGEMDOCK v2.1 T.E. (kcal/mol) | iGEMDOCK v2.1 VDW | iGEMDOCK v2.1 HB | iGEMDOCK v2.1 EI |
|---|---|---|---|---|---|---|---|---|---|
| 1 | Solanidine | −9.91 | 54.69 nM | −10.5 | 18.97 nM | −91.31 | −82.15 | −9.16 | 0 |
| 2 | Solasodine | −9.88 | 57.36 nM | −10.3 | 26.37 nM | −90.81 | −82.66 | −8.15 | 0 |
| 3 | Alpha-Solanine | −9.76 | 70.25 nM | −9.4 | 124.64 nM | −101.54 | −86.01 | −15.54 | 0 |
| 4 | Solasonine | −10.23 | 24.02 nM | −9.2 | 145.12 nM | −111.1 | −76.17 | −34.92 | 0 |
| 5 | Solamargine | −6.63 | 13.75 μM | −10.1 | 40.3 nM | −113.79 | −85.04 | −28.75 | 0 |
| 6 | Degalactotigonin | −1.90 | 20.08 mM | −9.5 | 85.29 nM | −122.96 | −83.1 | −39.86 | 0 |
| 7 | Nigrumnin-I | −5.12 | 70.59 μM | −9.3 | 135.12 nM | −115.93 | −73.68 | −42.25 | 0 |
| 8 | Uttroside B | −4.37 | 630.19 μM | −8.6 | 468.76 nM | −120.84 | −120.84 | −6.67 | 0 |

**C. Energy parameters with Quality Analysis (Molprobity) using three docking softwares**

**Human Thymosin beta-4**

| S. No | Phytoconstituent name | AutoDock v4.2.6 B.E. (kcal/mol) | AutoDock v4.2.6 $K_d$ | AutoDock vina B.E. (kcal/mol) | AutoDock vina $K_d$ | iGEMDOCK v2.1 T.E. (kcal/mol) | iGEMDOCK v2.1 VDW | iGEMDOCK v2.1 HB | iGEMDOCK v2.1 EI |
|---|---|---|---|---|---|---|---|---|---|
| 1 | Solanidine | −5.65 | 72.66 μM | −7.2 | 5.68 μM | −70.57 | −68.41 | −2.17 | 0 |
| 2 | Solasodine | −5.95 | 43.46 μM | −6.5 | 15.46 μM | −74.82 | −71.32 | −3.5 | 0 |
| 3 | Alpha-Solanine | −7.45 | 3.44 μM | −5.9 | 43.43 μM | −95.8 | −81.45 | −14.36 | 0 |
| 4 | Solasonine | −7.48 | 3.27 μM | −6.9 | 8.1 μM | −87.38 | −72.81 | −14.57 | 0 |
| 5 | Solamargine | −6.94 | 8.24 μM | −6.9 | 8.1 μM | −95.96 | −83.14 | −12.82 | 0 |
| 6 | Degalactotigonin | −6.1 | 33.65 μM | −6.1 | 33.18 μM | −96.65 | −72.86 | −23.79 | 0 |
| 7 | Nigrumnin-I | −7.78 | 1.99 μM | −5.8 | 52.13 μM | −101.76 | −80.58 | −21.18 | 0 |
| 8 | Uttroside B | −1.32 | 107.39 mM | −4.7 | 348.08 μM | −100.45 | −100.45 | 0 | 0 |

**Notes.**

⁻No binding detected.

**Table 2  Best docking poses of major components in *S. nigrum* with target protein(s).**

**A.**

| S. No. | Phytoconstituent name | Chemical class | Best docking pose with protein | Interacting amino acid(s) |
|---|---|---|---|---|
| 1 | Solamargine | Steroidal glycoalkaloid | Ezrin | Pro56, Thr57, Arg279,Trp217, Glu114, Tyr205, Lys233, His48, Ileu203,Lys233 |

**B.**

| | | | With quality analysis (Molprobity) using three docking softwares | | |
|---|---|---|---|---|---|
| | | | Human Coronin-1 | | |
| S. No | Phytoconstituent name | Chemical class | AutoDock v4.2.6 Interacting amino acid(s) | AutoDock vina Interacting amino acid(s) | iGEMDOCK v2.1 Interacting amino acid(s) |
| 2 | Solanidine | Steroidal aglycone | Ala41, Val42, Pro323, Pro277, Arg186, Phe275, Phe274, Asp183,Val227, Phe228, Pro273 | Arg186, Pro91, Pro277, Phe275, Val229, Phe228, Arg26, Pro323 | Pro91, Arg325, Ala41, Val42, Trp89, Cys90, Phe274, Tyr321, Pro323 |
| 3 | Solasodine | Steroidal aglycone | Asp183, Ala41, Phe274, Phe275, Cys48, Ser185, Val42, Gly320, Val227, Pro277 | Pro323, Lys324, Pro277, Tyr321, Phe274, Gly38 | Cys40, Pro91, Gly38, Ala41, Val42, Trp89, Cys90, Tyr321, Pro323 |
| 4 | Alpha-Solanine | Steroidal glycoalkaloid | Pro273, Val137, Phe274, Phe275, Ala41, Pro277, Val182, Val227, Ile136, Val4 | Thr351, Lys331, Ala349, Gln17, Pro347, Ala19, Gln23, Leu44, Thr67, Arg336 | Lys45, Leu65, Gly326, Gln17, Gln23, Leu64, Glu328, Arg336 |
| 5 | Solasonine | Steroidal glycoalkaloid | Ala88, Ala138, Val182, Ileu87, Asp186, Val137, Thr141, Arg186, Gly231, Pro323 | Ala19, Arg336, Glu328, Asp94, Pro398, Glu328, Leu327, Gly393, Arg325, Pro398, Lys345, Ser401, Lys400 | Gly38, Phe39, Cys40, Ileu87, Val182, Arg225, Asp36, Ser77, Tyr180, Arg196, Leu272, Pro273 |
| 6 | Uttroside B | Steroidal saponin | Tyr321, Phe274, Pro323, Val42, Ala41, Gly38, Ser37, Arg225, Asp186, Ileu87, Ala88, Trp89, Pro91, Thr141, Trp184 | Gly38, Ser37, Arg225, Thr321, Val42, Ala41 | Phe15, Gln17, Ala19, Gln23, Pro44, Leu64, Arg325, Gly326, Thr351, Lys355 |

**C.**

| | | | With quality analysis (Molprobity) using three docking softwares | | |
|---|---|---|---|---|---|
| | | | Human Thymosin beta-4 | | |
| S. No | Phytoconstituent name | Chemical class | AutoDock v4.2.6 Interacting amino acid(s) | AutoDock vina Interacting amino acid(s) | iGEMDOCK v2.1 Interacting amino acid(s) |
| 7 | Degalactotigonin | Steroidal saponin | Lys15, Leu18, Lys19, Thr21, Glu25, Gln24, Lys26 | Lys19, Lys20, Leu18, Thr21, Lys15, Phe15, Ileu10, Glu9 | Thr21, Glu22, Thr23, Gln24, Lys26, Pro28, Lys20 |
| 8 | Nigrumnin-I | Steroidal saponin | Asn27, Lys26, Glu25, Gln24, Thr23, Lys20, Leu18, Lys15 | Thr34, Ser31, Pro30, Leu29, Lys26, Glu25, Gln24 | Glu22, Gln24, Lys26, Lys20, Thr21, Thr23 |

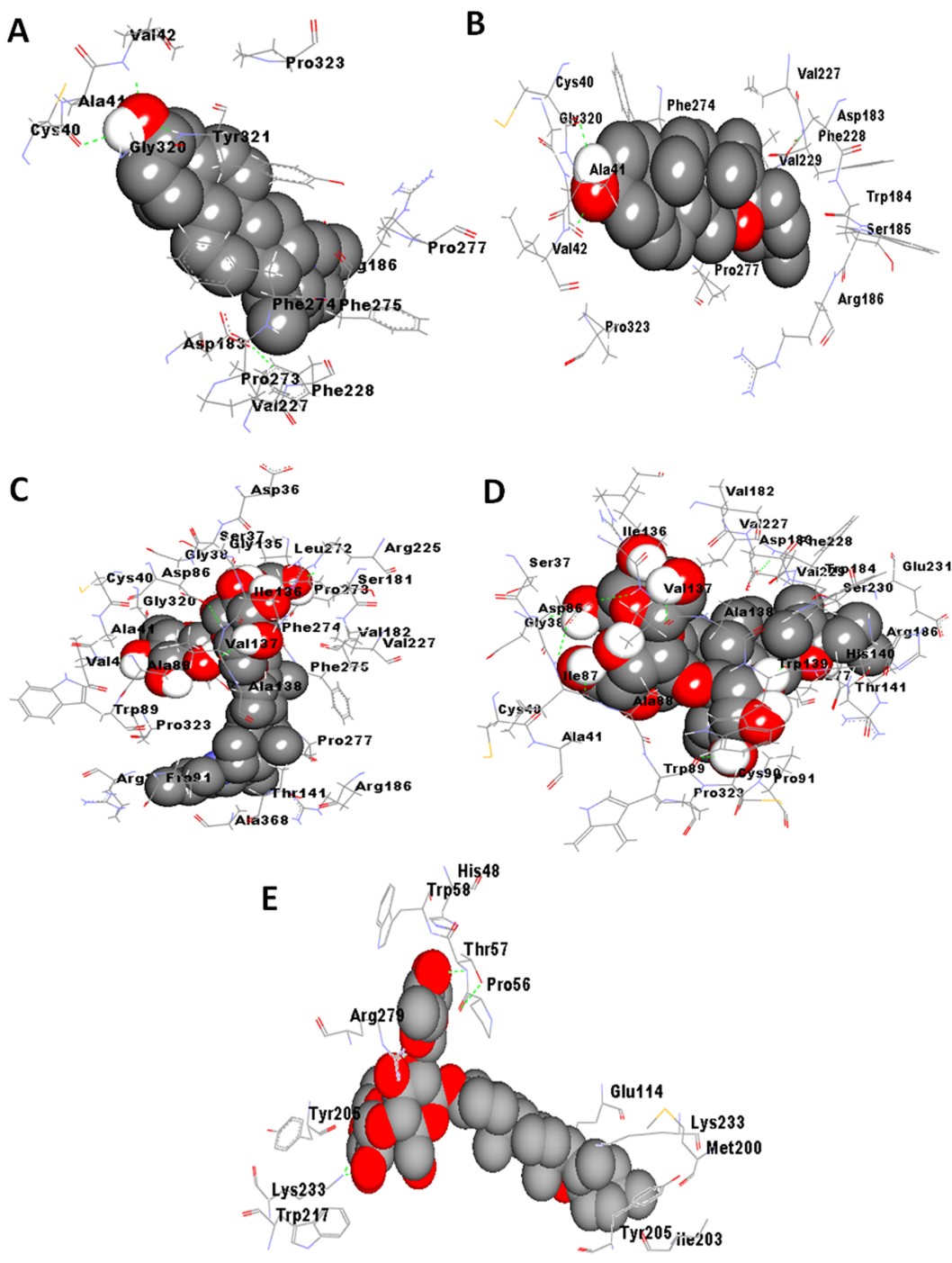

**Figure 2** Best docking poses for binding of steroidal alkaloids viz. (A) solanidine, (B) solasodine, (C) alpha-solanine, (D) solasonine with coronin-1A and (E) solamargine with ezrin. The poses were generated using AutoDock v4.2.6 (http://autodock.scripps.edu/).

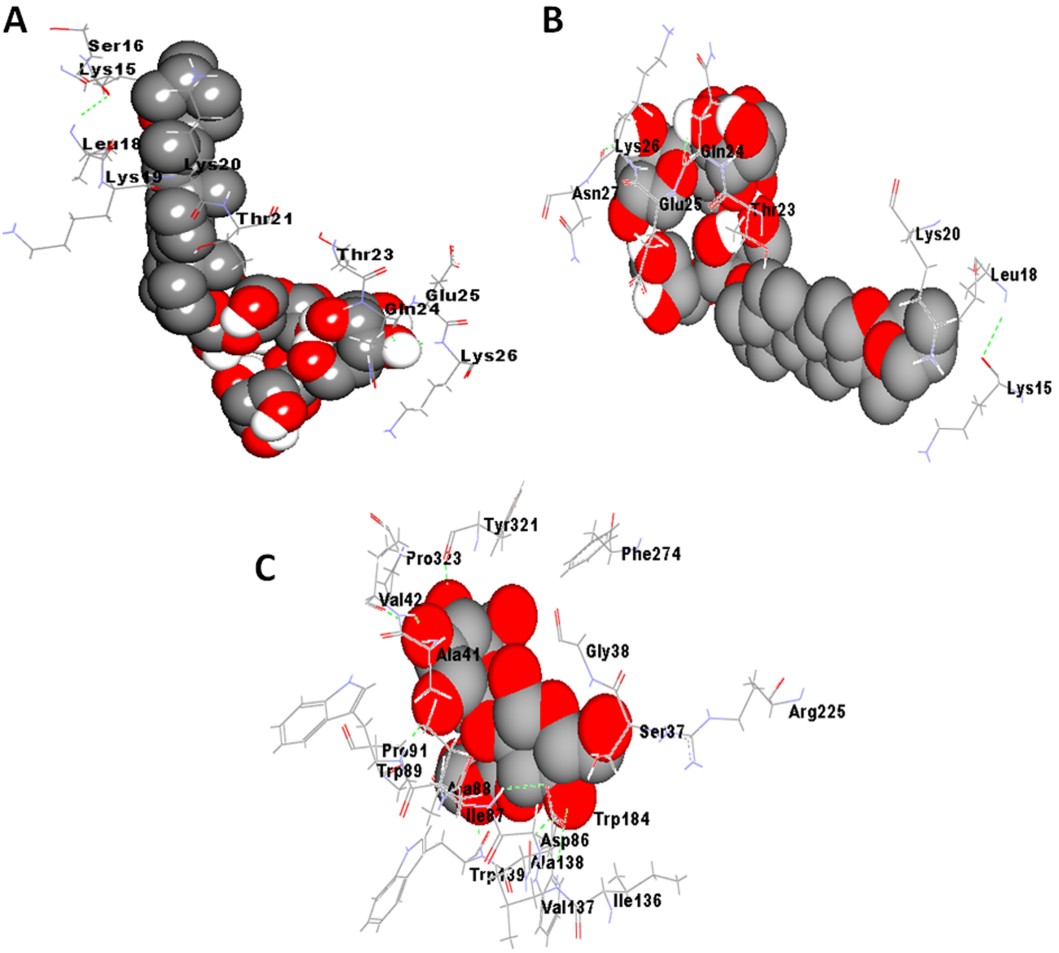

**Figure 3  Best docking poses for binding of steroidal saponins viz. (A) degalactotigonin, (B) nigrumnin-I with thymosin beta-4 and (C) uttroside B with coronin-1-A. The poses were generated using AutoDock v4.2.6 (http://autodock.scripps.edu/).**

variable degrees of binding to smaller physiological ligands like zinc ($K_d$ 139.32 mM), acetate ($K_d$ 6.92 mM) and sulfate (3.19 mM). This too, was found in agreement with previous studies (*Kammerer et al., 2005*). Thymosin β-4 showed some binding to calcium ($K_d$ 168.72 mM) as reported in literature (*Xue et al., 2014*). Ezrin displayed strong binding to PIP$_2$ ($K_d$ 8.5 µM) as reported previously (*Niggli et al., 1995*). It also had some affinity towards sulfate ($K_d$ 3.19 mM) as reported earlier (*Phang et al., 2016*). Cofilin also displayed binding to magnesium ($K_d$ 109.84 mM), in accordance to previous reports (*Hamill et al., 2016*). As is evident from Table 3, most of the studied proteins displayed potent binding to ATP (*Kambe et al., 1992*; *Szatmári et al., 2018*). However, the dissociation constant(s) of a majority of phytoconstituents from *S. nigrum* were found to be lower than those for ATP with respect to the studied target protein(s). Figure S6 depicts the best docking poses of cytoskeletal proteins with known physiological ligands. It is clear from the results that most of the studied phytocomponents, particularly solanidine, solasodine, alpha-solanine

**Table 3  Binding energies and dissociation constants ($K_d$) of some known physiological ligands with target protein(s).**

| S. No | Protein target | Physiological ligand | Experimental binding constants | AutoDock v4.2.6 B.E. (kcal/mol) | $K_d$ | AutoDock vina B.E. (kcal/mol) | $K_d$[a] | iGEMDOCK v2.1 T.E. (kcal/mol) | VDW | HB | EI |
|---|---|---|---|---|---|---|---|---|---|---|---|
| 1 | Vimentin | Acetate | – | −1.5 | 79.67 mM | −3.0 | 6.45 mM | −39.24 | −23.28 | −11.98 | −3.97 |
| | | ATP | 14.0 μM (*Esue et al., 2006*) | −1.26 | 119.56 mM | −7.3 | 4.0 μM | −107.26 | −53.35 | −45.43 | −8.47 |
| | | N-acetyl glucosamine | 55.7 μM (*Ise et al., 2017*) | −5.05 | 197.38 μM | −6.6 | 15.46 μM | −89.06 | −54.77 | −34.29 | 0 |
| 2 | Gelsolin | Calcium | 1.0 μM (*Lamb et al., 1993*) | −0.84 | 241.61 mM | −1.3 | 108.84 mM | – | – | – | – |
| | | ATP | 2.4 μM (*Szatmári et al., 2018*) | −2.31 | 20.28 mM | −6.8 | 9.50 μM | −117.01 | −72.39 | −40.43 | −4.19 |
| | | Phosphatidylinositol (4,5)-bisphosphate | 40.2 μM (*Lin et al., 1997*) | – | – | −5.9 | 43.43 μM | −113.81 | −84.67 | −27.49 | −1.65 |
| | | Phosphatidylinositol 4- monophosphate | ND | – | – | −7.2 | 4.39 μM | −103.32 | −68.52 | −29.74 | −5.06 |
| 3 | Villin | Acetate | ND | −2.24 | 22.69 mM | −2.9 | 6.0 mM | −36.47 | −16.07 | −20.02 | −0.2 |
| | | Sulfate | ND | −2.79 | 9.06 mM | −3.2 | 4.75 mM | −45.35 | −12.42 | −27.32 | −3.61 |
| | | ATP | ND | −4.1 | 989.87 μM | −7.4 | 3.75 μM | −100.82 | −51.19 | −41.62 | −8.01 |
| | | Phosphatidylinositol (4,5)-bisphosphate | 39.5 μM (*Kumar et al., 2004*) | – | – | −6.0 | 39.91 μM | −123.84 | −88.58 | −28.72 | −6.55 |
| | | Phosphatidylinositol 4- monophosphate | ND | – | – | −6.8 | 10.02 μM | −104.78 | −92.83 | −8.78 | −3.18 |
| 4 | Coronin-1A | Zinc | ND | −0.76 | 275.13 mM | −1.2 | 139.32 mM | – | – | – | – |
| | | ATP | ND | −3.68 | 2.02 mM | −8.3 | 869.1 nM | −108.04 | −83.02 | −29.21 | 4.2 |
| | | Acetate | ND | −2.64 | 11.69 mM | −2.9 | 6.92 mM | −57.3 | −28.34 | −24.71 | −4.26 |
| | | Sulfate | ND | −2.48 | 15.33 mM | −3.4 | 3.19 mM | −59.23 | −29.6 | −25.45 | −4.18 |
| | | Phosphatidylinositol (4,5)-bisphosphate | 0.64 μM[b] (*Olshina et al., 2015*) | – | – | −5.6 | 76.59 μM | −82.84 | −79.34 | −3.5 | 0 |
| | | Phosphatidylinositol 4- monophosphate | ND | – | – | −6.7 | 15.3 μM | −135.26 | −112.19 | −23.89 | 0.81 |
| 5 | Thymosin β-4 | ATP | 1.8-22 μM (*Carlier et al., 1993*) | −1.89 | 41.01 mM | −4.8 | 300.02 μM | −78.74 | −53.85 | −21.79 | −3.1 |
| | | Calcium | ND | −0.45 | 468.21 mM | −1.0 | 168.72 mM | – | – | – | – |
| | | Sulfate | ND | −6.68 | 12.74 μM | −3.4 | 3.19 mM | −57.55 | −17.29 | −31.39 | −8.88 |
| 6 | Ezrin | ATP | ND | −4.74 | 332.92 μM | −8.6 | 515.45 nM | −114.68 | −56.11 | −51.47 | −7.1 |
| | | Phosphatidylinositol (4,5)-bisphosphate | 52 nM (*Bosk et al., 2011*) | – | – | −6.9 | 8.5 μM | −122.25 | −78.09 | −37.01 | −7.15 |
| 7 | Cofilin | Magnesium | ND | −0.95 | 202.36 mM | −1.3 | 109.84 mM | – | – | – | – |
| | | ATP | 0.08 μM (*Chen & Pollard, 2011*) | – | – | −5.8 | 55.7 μM | −105.96 | −72.8 | −31.61 | −1.54 |

**Notes.**

[a]Dissociation constants obtained using AutoDock vina have been cited in the text on account of greater consistency to reported experimental values of the ligands for the respective target proteins.

[b]As determined for coronin from *Plasmodium falciparum*

−No significant binding detected.

ND, Experimental binding constants unknown/not determined.

and solasonine from *S. nigrum* showed stronger binding to the target proteins than their respective physiological ligands (Tables 1 and 3).

## PASS analysis: Lipinski's parameters

In general, an orally active drug should have no more than one violation of Lipinski's parameters otherwise its bioavailability is compromised. Compounds with MW <500 can be easily transported, diffused and absorbed. Number of rotatable bonds in compounds exhibiting biological activity should be <10, thus indicating their lesser molecular flexibility. TPSA is correlated with the hydrogen bonding of a molecule and can be used as a good indicator of the bioavailability of drug molecules. The property also characterizes the transport properties of a drug and should be well below the limit of <160 Å for biologically active compounds. Osiris tool identified the property based on summation of surface contributors of polar fragments (*Ertl, Rohde & Selzer, 2000*). The percentage of absorption calculated from TPSA should be above 50% to indicate good oral bioavailability (*Namachivayam, Raj & Kandakatla, 2014*). In the present study, the aglycones and their glyco-derivatives as well as saponins respectively had one and three violations of Lipinski's parameters (Table 4). This is an indication that there is a need to modify their structure–activity relationship (SAR) for improving their bioactivity and bioavailability. Of the standard drugs, doxorubicin HCl violated three of Lipinski's parameters, whereas tetracycline violated one (Table 4).

The proportion of non-hydrocarbon atoms among non-hydrogen atoms gives the heavy atoms proportion or 'R value' for a likely drug candidate. The best R value candidate drugs lie in the range of 0.05–0.50 (preferably 0.10–0.35). If a candidate drug possesses good drug-like properties and has large possibility to be developed into an approved drug, its heavy atoms count should be not more than its carbon atom count. Typically, for a compound to behave as an anticancer or antibacterial drug, the proportion of heavy atoms should be large. Conversely, oral drugs, and drug candidates for CNS and CVDs have smaller proportion of heavy atoms (*Mao et al., 2016*).

## Bioactivity score prediction and druglikeness

The bioactivity scores of all phytoconstituents are depicted in Table 5. As a general rule, greater is the bioactivity score, higher is the probability that the investigated compound would be biologically active. While both aglycones exhibited good bioactivity scores of more than 0.0 with respect to most of the receptor proteins and enzymes, most of their carbohydrate derivatives as well as saponins showed moderate bioactivity scores between −5.0 and 0.0, which clearly indicate that they possess such properties as are required for them to act as potential drugs with some modifications in their chemical structures. None of the phytoconstituents had bioactivity scores less than −5.0. As expected, doxorubicin HCl and tetracycline displayed bioactivity scores between −5 and >0 against most of the receptor proteins and enzymes (Table 5).

Druglikeness can be predicted by comparing structural features of compounds with structural features of marketed drugs. Molar lipophilicity as indicated by c Log P was ≤5 for all compounds except solanidine (c Log P = 5.21) showing their good permeability

**Table 4  PASS Analysis of major components in *S. nigrum* versus anticancer drug (doxorubicin HCl) and antibiotic (tetracycline) calculated by OSIRIS Property Explorer.**

| | | Physicochemical parameters | | | | | | | | |
|---|---|---|---|---|---|---|---|---|---|---|
| | | | | Lipinski's rule of 5 parameters | | | | | | |
| S. No. | Phytoconstituent name | % Absorption[a] (>50%) | Topological Polar Surface Area (Å)² (TPSA)[b] (<160 Å) | MW (<500) | c log P[c] (<5) | Heavy atom count | Hydrogen Bond Donors (nOHNH) ($\leq$5) | Hydrogen Bond Acceptors (nON) ($\leq$10) | Number of Rotatable bonds ($\leq$10) | Lipinski's violation |
| 1 | Solanidine | 100.9 | 23.47 | 397.64 | 5.21 | 29 | 1 | 2 | 0 | 1 |
| 2 | Solasodine | 94.7 | 41.49 | 413.65 | 4.74 | 30 | 2 | 3 | 0 | 1 |
| 3 | Alpha-Solanine | 28.97 | 240.69 | 868.07 | 0.62 | 61 | 9 | 16 | 8 | 3 |
| 4 | Solasonine | 19.74 | 258.72 | 884.07 | 0.15 | 62 | 10 | 17 | 8 | 3 |
| 5 | Solamargine | 26.72 | 238.48 | 868.08 | 1.08 | 61 | 9 | 16 | 7 | 3 |
| 6 | Degalactotigonin | −6.6 | 335.08 | 1,035.18 | −1.76 | 72 | 12 | 22 | 11 | 3 |
| 7 | Uttroside B | −44.68 | 445.44 | 1,215.34 | −3.96 | 84 | 17 | 28 | 18 | 3 |
| 8 | Nigrumnin-I | −19.94 | 373.75 | 1,151.29 | −2.07 | 80 | 13 | 25 | 12 | 3 |
| 9 | Doxorubicin HCl | 37.90 | 206.08 | 579.83 | 0.18 | 40 | 7 | 12 | 5 | 3 |
| 10 | Tetracycline | 46.34 | 181.61 | 444.44 | −1.26 | 32 | 7 | 10 | 2 | 1 |

Notes.
[a] Percentage Absorption was calculated as: % Absorption = 109 − [0.345×Topological Polar Surface Area].
[b] Topological polar surface area (defined as a sum of surfaces of polar atoms in a molecule).
[c] Logarithm of compound partition coefficient between n-octanol and water.

**Table 5  Bioactivity score and Druglikeness of major components in *S. nigrum* versus anticancer drug (doxorubicin HCl) and antibiotic (tetracycline) calculated by Molinspiration software.**

| | | Parameters of Bioactivity score | | | | | | |
|---|---|---|---|---|---|---|---|---|
| S. No. | Phytoconstituent name | GPCR ligand | Ion channel modulator | Kinase inhibitor | Nuclear receptor ligand | Protease inhibitor | Enzyme inhibitor | Druglikeness |
| 1 | Solanidine | 0.39 | 0.05 | −0.44 | 0.56 | 0.21 | 0.57 | 3.31 |
| 2 | Solasodine | 0.24 | −0.17 | −0.66 | 0.36 | 0.01 | 0.60 | 2.67 |
| 3 | Alpha-Solanine | −2.38 | −3.42 | −3.44 | −3.13 | −1.82 | −2.61 | −0.22 |
| 4 | Solasonine | −2.65 | −3.55 | −3.57 | −3.35 | −2.12 | −2.74 | −0.97 |
| 5 | Solamargine | −2.45 | −3.51 | −3.52 | −3.22 | −1.92 | −2.59 | 3.16 |
| 6 | Degalactotigonin | −3.64 | −3.75 | −3.80 | −3.75 | −3.55 | −3.60 | −15.55 |
| 7 | Nigrumnin-I | −3.79 | −3.85 | −3.89 | −3.86 | −3.74 | −3.75 | −3.71 |
| 8 | Uttroside B | −3.83 | −3.88 | −3.92 | −3.89 | −3.80 | −3.80 | −14.16 |
| 9 | Doxorubicin HCl | 0.20 | −0.20 | −0.07 | 0.32 | 0.67 | 0.66 | 6.65 |
| 10 | Tetracycline | −0.15 | −0.24 | −0.53 | −0.09 | −0.04 | 0.52 | 5.57 |

across cell membranes. Positive values of druglikeness were calculated for *S. nigrum* phytoconstituents solanidine, solasodine and solamargine which indicated that these compounds contain fragments that are present in marketed drugs. As expected, both the reference drugs exhibited positive scores for druglikeness (Table 5).

**Table 6** Toxicity calculations of *S. nigrum* phytoconstituents versus anticancer drug (doxorubicin HCl) and antibiotic (tetracycline) calculated by Osiris property explorer.

| S.No. | Phytoconstituent name | Mutagenic | Tumorigenic | Reproductive effective | Irritant |
|-------|----------------------|-----------|-------------|------------------------|----------|
| 1 | Solanidine | None | None | Mild | None |
| 2 | Solasodine | None | None | Mild | None |
| 3 | Alpha-Solanine | None | None | Mild | None |
| 4 | Solasonine | None | None | None | None |
| 5 | Solamargine | None | None | Mild | None |
| 6 | Degalactotigonin | None | None | None | None |
| 7 | Uttroside B | None | None | None | None |
| 8 | Nigrumnin-I | None | None | None | None |
| 9 | Doxorubicin HCl | None | None | None | High |
| 10 | Tetracycline | None | None | High | High |

**Notes.**

None — (green)

Mild — (yellow)

High — (red)

## Toxicity potential evaluation

The toxicity risk assessment is essential to avoid unsuitable substances for further drug screening, if they are predicted to have adverse side effects on biological system (*Balakrishnan, Raj & Kandakatla, 2015*). The phytocompounds were screened for any mutagenic, tumorigenic, irritant and reproductive toxicity risks. The toxicity risk of phytoconstituents versus anticancer drug doxorubicin HCl and antibotiic tetracycline was predicted by means of pre-computed set of structural fragments by Osiris Property Explorer. The program predicts the toxicity potential of the compounds on the basis of similarities of the investigated compounds with the studied compounds present in its database *in vitro* and *in vivo* (*Sander, 2001*). The results of the prediction were color coded and are represented in Table 6 (*Jagadish, Soni & Verma, 2013*). Properties shown in red indicate high risk of undesired effects while a green color predicts conformation to drug-like behavior and compatibility (*Husain et al., 2016*). Unlike synthetic drugs doxorubicin and tetracycline, which were shown to have some potent irritant effects and adverse effects on the reproductive system (Table 6), the naturally occurring glycoalkaloids and saponins were predicted to be safe with no toxicity, tumorigenicity or irritant effect(s). Only solanidine, solasodine, alpha-solanine and solamargine were predicted to have only a mild to low effect on the reproductive system.

## Principle Component Analysis (PCA)

PCA was performed on three most variable properties viz. TPSA, % Absorption and MW using linear correlation (Figs. 4A, 4B, Tables 4 and 7). As mentioned earlier, except for aglycones solanidine and solasodine, MWs of all other studied phytoconstituents were >500, thus an impediment can be predicted in the transport, diffusion and absorption of

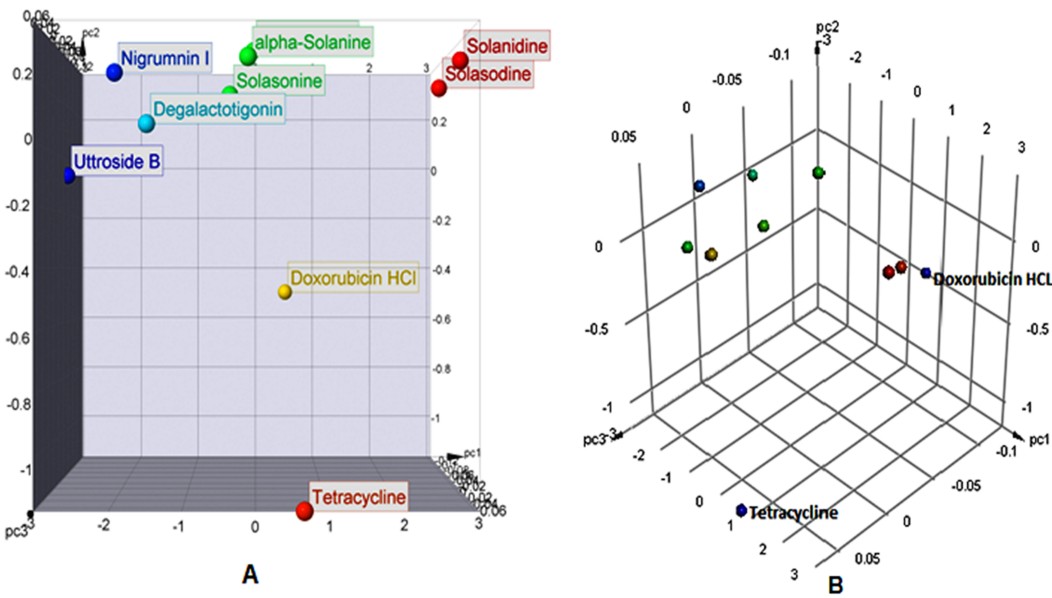

**Figure 4** PCA of physicochemical properties of selected phytoconstituents from *S. nigrum* versus reference drugs. (A) Scatter plot and (B) 3-D point plot.

these compounds. Related properties viz. % absorption and TPSA also reflect the need for structural modification/alteration of these compounds for better SAR.

## Structure–activity relationship

Analysis of the results suggested that the number, type and location of sugar or the substitution of hydroxyl group on steroidal alkaloid backbone had an effect on the activity (Fig. 5). Alpha-solanine (Fig. 5C) has a solanidane (Fig. 5A) type of backbone formed by an indolizidine ring where tertiary nitrogen connects the two rings and a solatriose type of sugar moiety (Fig. 5I). The spirosolane type of aglycone comprising of a tetrahydrofuran and piperidine spiro-linked bicyclic system with an oxa-azaspirodecane structure (as in solasonine) is more chemically reactive than solanidane aglycone (B). As is evident from Figs. 5A–5B, and Figs. 5I–5J solasonine (Fig. 5D) and solamargine (Fig. 5E) have the same steroid aglycone (solasodine type) but different sugars units (solatriose and chacotriose) respectively. Solasonine (Fig. 5D) and solamargine (Fig. 5E) contain α-L-rhamnopyranose located at the C-2 of β-D-glucose or galactose and exhibited strong binding (low $K_d$ values) against all studied proteins of cytoskeletal framework (Table 1). The rhamnose moiety at C-2 of the trisaccharide moiety appeared to be essential for alpha-solanine, solasonine and solamargine to display anticancer activity. A dramatic decrease in binding to the selected targeted proteins was observed (higher $K_d$ values) with degalactotigonin (F) due to the absence of this monosaccharide (Table 1) (*Milner et al., 2011*; *Chang et al., 1998*). Nigrumnin-I (Fig. 5G) by virtue of it containing rhamnopyranose at C-2 exhibited higher activity (lower $K_d$ values) against villin, ezrin, coronin-1A and thymosin beta-4.

**Table 7** Bravais-Pearson (linear correlation) coefficient of physicochemical properties of selected phytoconstituents from *S. nigrum* versus reference drugs (doxorubicin HCl and tetracycline).

| Property | | 1 | 2 | 3 | 4 | 5 | 6 | 7 |
|---|---|---|---|---|---|---|---|---|
| % Abs | 1 | | −1 | −0.939 | 0.937 | 1 | −6.67E−04 | 0.0338 |
| TPSA | 2 | −1 | | 0.941 | −0.936 | −1 | 0.00476 | −0.0255 |
| MW | 3 | −0.939 | 0.941 | | −0.768 | −0.941 | 0.338 | 0.032 |
| clogP | 4 | 0.937 | −0.936 | −0.768 | | 0.938 | 0.344 | −0.031 |
| pc1 | 5 | 1 | −1 | −0.941 | 0.938 | | −1.12E−09 | 2.28E−09 |
| pc2 | 6 | −6.67E−04 | 0.00476 | 0.338 | 0.344 | −1.12E−09 | | −6.56E−09 |
| pc3 | 7 | 0.0338 | −0.0255 | 0.032 | −0.031 | 2.28E−09 | −6.56E−09 | |

Solasodine (B), the aglycone of solasonine and solamargine has also been known to exhibit potent antitumor activity (*Liu, 2005*; *Liu, 2004*; *Trouillas et al., 2005*). In the present study, glycoalkaloids solasonine and solamargine exhibited potent binding to vimentin, gelsolin, villin, ezrin, coronin 1-A and thymosin-beta 4 (Table 1) indicating that the trisaccharide side chain is essential for activity (*Lee et al., 2004a*; *Lee et al., 2004b*; *Chang et al., 1998*). Structural modification of the aglycone at the 3-OH position has been found to result in a dramatic increase in antiproliferative potential of solasodine in PC-3 cell lines (*Zha et al., 2010*). Sulfation of the sugar chain has not been found to cause an increase in the activity of glycoalkaloids but the nitrogen atom in solasodine is critical for activity and any substitution at these positions has been found to result in complete loss of activity (*Zha et al., 2010*).

A number of plant extracts containing the spirosolane glycoalkaloids solamargine and solasonine and their aglycone solasodine have been examined for their anticancer activity. As mentioned above, the α-L-rhamnopyranose sugar at C-2 is critical to activity as β2-solamargine (in which the rhamnose sugar is hydrolyzed) and solasodine (the aglycone, which does have the sugar unit) have been found to be either inactive or with negligible activity (*Bhutani et al., 2010*; *Esteves-Souza et al., 2002*). In our study too, solamargine displayed lesser activity (higher $K_d$ value) towards vimentin, gelsolin and ezrin as compared to solasonine and the parent compound solasodine (Table 1).

Moreover, it was found that the steroidal alkaloids with trisaccharides (alpha-solanine, solasonine and solamargine) had higher activity (lower $K_d$ values) than steroidal glycosides having saccharide chains with more than three sugar residues (degalactotigonin, nigrumnin-I and uttroside B). This might be due to the more bulky nature of the steroidal glycosides that may pose a steric hindrance to their solubility and transport kinetics. Location and number of rhamnose moieties also appears to play an important role in activity (solamargine).

Of the various mechanisms by which glycoalkaloids may exert their anticancer effect, perhaps the most widely studied is through the suppression of the cell cycle at G2/M phase *via* inhibition of expression of Bcl-2 protein, thereby causing apoptosis (*Ji et al., 2008*; *Chowański et al., 2016*). Though the glycoalkaloids are more selective towards cancer cells, they have also been shown to be toxic to a few normal human cell lines *in vitro* (*Friedman et al., 2005*) and non-toxic when fed to human volunteers at about 200 mg/kg (*Mensinga et*

**Figure 5** Structure of the major glycoalkaloids of *S. nigrum*: aglycones solanidine (A) and solasodine (B) and their glycosylated derivatives α-solanine (C), solasonine (D) and solamargine (E); their constituent carbohydrate moieties (I–J) and steroidal saponins degalactototigonin (F), nigrumnin-I (G) and uttroside B (H). Redrawn from *Chowański et al. (2016)*.

*al., 2005*). These controversies and conflicting reports regarding selectivity of glycoalkaloids towards cancer and normal cells have so far restricted their widespread therapeutic use (*Lee et al., 2004a*; *Lee et al., 2004b*; *Friedman et al., 2005*).

## DISCUSSION

SGAs constitute a group of glycosidic derivatives of nitrogen-containing steroids that are produced as secondary metabolites in many plant species, especially those belonging to Solanaceae and Liliaceae families (*Roddick, 1996*). They consist of a C27 cholestane skeleton (aglycone) to which a carbohydrate moiety of one to five monosaccharides is attached at the 3-OH position of the aglycone (*Roddick, 1996*; *Van Gelder & Scheffer, 1991*). The carbohydrate moiety consists of diverse combinations of D-glucose, D-galactose, D-xylose and L-rhamnose (*Roddick, 1996*). Since the nitrogen atom is acquired though transition to complete the heterocycle, these compounds are pseudo-alkaloids (isoprenoid alkaloids). The structural diversity in plant glycoalkaloids is attributed mainly to skeletal backbone of the aglycone structure (*Friedman & McDonald, 1997*) which is broadly divided into two main types viz. the spirosolane type of aglycone is solasodine, whereas solanidane type is solanidine. The nitrogen in aglycones can occur as primary (free or methylated), secondary (ring-closed) or tertiary (in two rings) and is responsible for influencing the chemical nature and property of the aglycone depending upon its substitution (*Dinan, Harmatha & Lafont, 2001*). Double bonds and OH groups occur at various positions resulting in further diversity of structures (*Friedman & McDonald, 1997*). The major glycoalkaloids found in *S. nigrum* are solanine, solasonine and solamargine. Alpha-solanine (C) consists of the aglycone solanidine (A) to which a trisaccharide moiety solatriose, is attached (Figs. 5A, 5I). Solasonine (D) and solamargine (E) share a common aglycone, solasodine (B), to which a trisaccharide moiety, solatriose (solasonine) or chacotriose (solamargine) is attached (Figs. 5I, 5J).

Though there a numerous reports on the role and *in vitro* effect of alkaloids at cellular levels viz. disruption of biological membranes and cellular metabolism, there are few reports pertaining to usefulness of these compounds in cancer therapy (*Dinan, Harmatha & Lafont, 2001*; *Mohanan & Devi, 1996*). There is scanty information regarding the antiproliferative potential of solasonine but it has been reported to act in a concerted manner along with solamargine *in vitro* (*Friedman, 2015*). In our ongoing study on evaluation of anticancer activity of alcoholic extract(s) of *S. nigrum* against human cancer and normal cell lines, the leaf extract of *S. nigrum* has been found to display potent cytotoxic activity against HepG2 and MDA cells ($IC_{50}$ values approx. 20 µg/mL) whereas no appreciable cytotoxicity was observed against the normal cells (*Ahmad et al., 2017*). Curaderm[BEC5], a herbal preparation, containing a combination of solasonine and solamargine, is commercially available for treatment of skin cancers. Furthermore, for treatment of internal cancers, an intravenous preparation of the same solasodine glycoalkaloids has entered phase II clinical trials. It has also been found that a rapid regression of solid tumors occurs when these compounds are administered by intralesion injection into solid tumors (*Millward et al., 2005*; *Cham, Daunter & Evans, 1991*; *Cham, 1994*; *Cham, 2008*; *Cham & Daunter, 1990*;

*Daunter & Cham, 1990*; *Cham, Gilliver & Wilson, 1987*; *Punjabi et al., 2008*). Therefore, the use of solasodine and its derivative glycoalkaoids, along with chemotherapeutic drug cisplatin may serve as an effective combination regimen especially in cisplatin resistant breast cancer (*Shiu et al., 2007*; *Pietras et al., 1999*).

Triterpene and steroidal glycosides commonly referred to as saponins are another major group of phytoconstituents naturally occurring in plants belonging to Solanaceae. Like alkaloids, saponins consist of an aglycone (glycoside-free) part known as sapogenin. The number of saccharide chains attached to the sapogenin/aglycone core is variable and can be linear or branched with D-glucose and D-galactose being the most common components of the chains (*Hostettmann & Marston, 1995*). The aglycone is lipophilic and can be a structurally diverse polycyclic organic structure with C10 terpene units forming a C30 triterpene skeleton (*Suzuki et al., 2002*; *Foerster, 2017*) or subsequent structural modification to produce a C27 steroidal skeleton (*Hostettmann & Marston, 1995*). Those saponins that have a steroidal backbone are known as saraponins (*Cornell University, 2008*). The aglycone part may also contain nitrogen; such saponins exhibit chemical and pharmacologic properties similar to alkaloids. The steroidal aglycone can be spirostanol type bearing a sugar chain linked to C-3 or furostanols type carrying a sugar chain at C-3 and a D-glucose residue at C-26. The spirostanol saponins degalactotigonin (F) and nigrumnin-I (G); (Fig. 5) usually contain diosgenin, tigogenin and gitogenin as aglycones (*De Combarieu et al., 2003*; *Ikeda, Tsumagari & Nohara, 2000*) whereas furostanols (uttroside B (H); Fig. 5) usually contain a steroidal furanose ring (*Nath et al., 2016*).

Saponins have been reported to exhibit diverse pharmacological activities owing to their structural diversity (*Vincken, De Groot & Gruppen, 2007*). Out of their numerous biological activities, their potent antitumor effect has been widely reported and extensively studied (*Jain et al., 2011*; *Zhou et al., 2006*; *Milner et al., 2011*; *Ikeda, Tsumagari & Nohara, 2000*; *Podolak, Galanty & Sobolewska, 2010*; *Hu et al., 1999*; *Jin, Zhang & Yang, 2004*).

In the present study, additional prospective targets of solasonine and solamargine have been elucidated using molecular docking. It was found that both the spirosolane SGAs showed potent binding to the newly proposed molecular targets (Fig. 2, Table 1). The mechanism of action may be related to structural alteration/modification of cytoskeletal network, actin stabilization/destabilization, intermediate filament assembly/disassembly, and EMT inhibition along with induction of apoptosis.

## CONCLUSION

This study is the first of its kind using *in silico* docking to explore the specific molecular targets to which these compounds may bind prospectively *in vivo* and thereby exert their anticancer or antibacterial activity. The study revealed some new targets for *S. nigrum* phytoconstituents, other than those already known. Solanidine, solasodine, alpha-solanine, solasonine and uttroside B showed potent binding to human coronin-1A, whereas degalactotigonin and nigrumnin-I showed strongest binding to human *thymosin beta-4*. These phytoconsituents also showed strong bindings to other selected target proteins as well, albeit to a lower extent as compared to these two proteins. However, there is a need

for modification in the structures(s) of the naturally occurring constituents in order to meet certain absorption, distribution, metabolism, excretion, toxicology (ADMET) and solubility criteria set for potential drug-candidates. SGAs from *S. nigrum* with three sugar units and α-L-rhamnopyranose at C-2 or a hydroxyl group on the steroidal backbone may serve as potential anticancer candidates if studied and evaluated further in terms of their structure–activity relationships as well as their binding kinetics to certain cytoskeletal proteins. This may pave a way for discovery of other prospective targets for this important class of phytoconstituents apart from those already known.

## Abbreviations

| | |
|---|---|
| **S. nigrum** | *Solanum nigrum* |
| **ABP** | Actin Binding Protein |
| **PASS** | Prediction of Activity Spectra for Substances |
| **PCA** | Principle Component Analysis |
| **SAR** | Structure-Activity Relationship |

### Funding
The author received no funding for this work.

### Competing Interests
The author declares there are no competing interests.

### Author Contributions
- Rumana Ahmad conceived and designed the *in silico* experiments, performed the experiments, analyzed the data, contributed analysis tools, prepared figures and tables, authored and reviewed drafts of the paper as well as approved the final draft.

### Data Availability
The 3D crystal structures of prospective receptors/protein targets of phytoconstituents were downloaded from Protein Data Bank (http://www.rcsb.org/pdb) in PDB format. Direct links to the data used from this site are as follows:

https://www.rcsb.org/structure/1GK4, https://www.rcsb.org/structure/3FFN, https://www.rcsb.org/structure/3FG7, https://www.rcsb.org/structure/4RMA, https://www.rcsb.org/structure/4BAX.

### Supplemental Information
Supplemental information for this article can be found online at http://dx.doi.org/10.7717/peerj.6012#supplemental-information.

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
