# Peer review of "Steroidal glycoalkaloids from Solanum nigrum target cytoskeletal proteins: an in silico analysis"

_PeerJ, doi:10.7717/peerj.6012_

## Round 0.1 · original submission · Major Revisions

Please address the issues identified by our reviewers.

Additional review-level comments by the editor:

- No quality analysis of the homology models was performed (e.g. as avauilable online with Molprobity and other tools). Therefore, the reliability of the data obtained for COR1A and TYB4 cannot be ascertained.

- Table 6 must state the identity of the "reference drugs"

- The binding energies of the phytoconstituents must be compared to the energies predicted by the same docking procedure for experimentally-confirmed ligands.

- SN should be replaced by S. nigrum throughout, for ease of reading.

·

Basic reporting

- In several lines, for example, 12 and 51, has been used an ambiguous expression, which needs to change and explain more details about another reported works. At the same time, some lines need commas (16-18).
- The cited references in this work are sufficient, inclusive I consider that which are exhaustive and can be selected fewer references.
- Figure 2 presents shorter labels of the reside; Figure 4 present structures with a different format, which is necessary homologate in the same format and depict a figure with a better resolution.
- This work is self-contained with relevant results to hypotheses, although the work needs most conscious results and diminish the excessive background.

Experimental design

- The work is an original primary research within aims and scope of the journal.
- The author defines, implicitly, the research questions, which have been well defined, based on the kind of interactions presented between the cytoskeletal proteins and steroidal glycoalkaloids and can be relevant to design some compounds directed to these targets.
- About the rigorous research, the homology model depicted in Figure 1, which show the modeled structure of TYB4, is very simple, and cannot homologate a complete protein, therefore it is not a model obtained with a rigorous research.
- The methodology about the optimization of the ligand structures is incomplete (lines 177-178), due is only reported the employed software, but not the level of theory. At the same time, in lines 180-186, the refinement of the selected proteins is mentioned, but the software is not cited.

Validity of the findings

- The obtained results of this work present a novelty, but one of the homology models, TYB4, it is very simple and cannot represent the basic resides in a whole protein, for this reason, is necessary to obtain a better homology model and replicate the assays on this new model.
- The data is robust and controlled, the performed assays are sufficient to justify the results.
- The work conclusion needs to highlight more aspects of this work, due do not link completely the original research question with the results.

Additional comments

The author performs an in silico analysis of the interactions and pharmacological effects of them, about some steroidal glycoalkaloids from solanum nigrum with two cytoskeletal proteins. In general, the background is a lot of large and need some actual references, but justify the work done.
About the homology models, is clear the used methodology, but the percent of similarity and another measured parameters in a homology model, is not the only way to select the better model, at the same time is necessary to visualize the large and conformation of the protein, due to the possible allosteric interaction in the target surface, which is not the aim of this work, but is part of the obtaining models.

·

Basic reporting

Author presents an interesting molecular docking study of phytoconstituents against cytoskeletal proteins. Generally, the paper is well written, the Introduction is well laid out as to provide the readers with a general overview of the research question presented herein. However, there were some sentences that had grammatical errors and thus, it is suggested that the authors re-check and re-proof the manuscript for such language problems.

There are specific points for improvements as follows:
- Lines 137-138: The paragraph about Villin is too short. More details should be added here so that it would be of comparable depth and length as the other ABPs.
- Lines 164-168: It is suggested to delete this section as it does not any new information and is redundant with what is to come in the subsequent sub-sections.
- Line 176: Please correct "chEMBL" to "ChEMBL"
- Line 176: Please correct "Pubchem" to "PubChem"
- Line 178: Please correct "Auto Dock Tools" to "AutoDockTools"
- Line 181, 189-190: Please correct "protein data bank" to "Protein Data Bank"
- Line 208: Please correct "I-tasser" to "I-TASSER"
- Line 227: This section heading seems a bit confusing since the contents of this section primarily discusses mostly about Lipinski's Ro5. It is recommended to rename "PASS Analysis: Lipinski’s parameters" to "Lipinski’s rule-of-five". Please also integrate contents from Lines 243-247 to this section.
Then, please add a new section called "PASS analysis" and also provide more extensive details.
- Line 325: Please rename "SAR" to "Structure-activity relationship"
- Line 367: Correct "there have also been shown" to "they have also been shown"
- Figure 4: All chemical structures shown in this figure seemed to each be obtained from different sources since it is heterogeneous. It is recommended that the authors redraw all structures using the same software using the same ACS style (refer to https://chemtips.wordpress.com/2012/11/19/drawing-chemicals/).

Experimental design

- Generally, the experimental design was well laid out. However, it would be nice if the author could provide a workflow figure that summarizes all the components of this study visually.
- In order to facilitate reproducible research, it is advised that the author also provide the predicted protein structures (as a PDB file) as a Supplementary File. All Binding pose of chemical structures (in MOL file format) should also be provided as Supplementary Files.
- It is advised that the author should discuss about the validity and reliability of the predicted protein structures and also show relevant Ramachandran plots as well as other relevant metrics.

Validity of the findings

This study reports interesting insights gained on the potential bioactivity of the investigated phytoconstituents as well as the binding modes against their target proteins. Results seemed to be experimentally sound.

Additional comments

Authors presented a well laid out study and the insights provided herein are expected to serve as useful guidelines for further investigations. There are specific comments on how to improve the manuscript as provided above. The author is highly encouraged to share all PDB files of the predicted protein structures as well as share the binding poses as MOL files for all investigated ligands as Supplements.

Reviewer 3 ·

Basic reporting

no comment

Experimental design

no comment

Validity of the findings

no comment

Additional comments

The paper of Rumana Ahmad, deals with the study of Molecular docking analysis of Solanumnigrum alkaloids to cytoskeletal proteins Steroidal glycoalkaloids from Solanumnigrum target cytoskeletal proteins: An in silico analysis
The cytoskeletal target(s) for SN phyto constituents are identified by an in silico analysis. The results of molecular docking, SAR analysis, physicochemical properties and Toxicity prediction methods are appropriate. In my opinion the paper is suitable for publication in Journal of Peer j. Author has to mention the admissible ratio of physicochemical properties. In figure -2 the atom of substrate which binds to the enzyme is not visible.
I recommende this article may be suitable for publication in your esteemed journal.

---

## Round 0.2 · Major Revisions

Although I appreciate the care you have taken with your revision, there are still important issues you have not addressed:

a) Although the introduction is quite long and describes the diffferent proteins tested, it conspicuously fails to state why one would expect any of the tested ligands to be active against cytoskeletal proteins.

b) I still cannot understand the rationale behind your choice of "reference drugs": doxorubicin binds to DNA and tetracycline binds to the ribosome. There is no reason to expect them to bind to cytoskeletal proteins, nor to think that they provide a suitable control to identify cytoskeletal protein-binding compounds.

c) Comparisons with the physiological ligands are still missing. These must be provided to ascertain whether the docking energies you found mean a strong enough binding. I will not be able to accept this manuscript unless those computations are provided.


Other issues:

- Molprobity refinement mostly optimizes Asn/Gln rotamers and His protonation patterns. It is therefore not at all surprising that the binding energies post-refinement are almost identical to the original ones. A discussion of the few exceptions to this pattern might be useful (though only if they correspond to stronger binding than observed with the physiological ligands)

- Ramachandran plots and Molprobity tables should be moved to Supporting Information.

- Authors are not required to profusely than reviewers in every comment. Although I understand that different cultures have varying degrees of formality and ways to address power differentials (such as those between authors and reviewers/editors), excessive deference in the language used towards reviewers/editors may unwittingly be perceived (by people of a Western background) as disingenuous and insincere.

- Sequence similarity returned by Swiss Model is a sui generis metric quite unlike what people have in mind when they say "sequence similarity": instead of being the percentage of aminoacids which are identical/similar between target and template, it is a measure of the "penalty cost" of mutating one into the other using a BLOSUM62 matrix. Since this metric is extremely non-intuitive, I suggest you drop it from your report.

---

## Round 0.3 · Major Revisions

Before I relay your manuscript to the reviewers, I would advise you to include in the main text the research rationale you exposed in your rebuttal, as I believe that would make the Introduction flow more naturally.
In lines 355-362, it is not immediately clear that the quoted references are the experimental reports of the binding to physiological ligands. Please adjust the text accordingly.

Some additional effort is needed to obtain an estimate of the binding constants fot the phosphatidyl inositol compounds. I understand that the large number of torsions in their acyl chains make the conformation search unwieldy, but IF those acyl chains play no role in the binding (which you may confirm by checking if there are no extended exposed hydrophobic surfaces on the target protein) you might either try to freeze the acyl chains in their extended conformations or else replacing both acyl groups with acetyl groups, thereby dramatically cutting the number of torsions.

I am also surprised by the poor numbers you got for ATP binding. Can you show the experimental values in the table for ease of comparison?

---

## Round 0.4 · Minor Revisions

Both reviewers have deemed your manuscript ready for publication and I am ready to accept your manuscript. I have only a few minor things to ask from you:
A) Include the experimental binding constants of physiological ligands in table 3
B) In tables 1 and 2, for thymosin and coronin, include ONLY the data after Molprobity quality improvement
C) Figure 2 seems to be distorted horizontally. Please correct the distortion and divide it in 2 figures
D) In figure 4, solanidine (panel A) is missing a nitrogen atom. Solasodine (panel B) misses a double bond and the O atom in its furane ring seems to be drawn on top of a carbon atom. Please remove the repeated "solanidine, R-H" text from the caption of panel B and "alpha-solanine, R=solatriose" from the caption of panel A. Panel H seems to be distorted vertically.
E) Please provide the pdb structures of the complexes as Supporting Information.

·

Basic reporting

The article has been written better than the first version, presenting a clear and unambiguous, English.
The references are sufficient, and the article structure, figures, tables are more clear; the article presents self-contained results.

Experimental design

The article presents an original research within the aims and the scope of the journal, with a rigorous investigation.

The method described has been improved.

Validity of the findings

Data is robust and controlled. At the same time, conclusions are well stated, linked to the original research question and limited to supporting results.

Additional comments

The article can be published in the current form

·

Basic reporting

Authors have addressed all comments raised in the previous peer review comments.

Experimental design

Methods seem methodologically sound.

Validity of the findings

Reliability of findings have been addressed.

Additional comments

It is deemed that the manuscript is now ready for publication.

---

## Round 0.5 · accepted · Accept

I think your manuscript has become much stronger throughout the peer-review process and I am happy to accept it for publication in PeerJ.

#